# Economic burden of brain metastases in non-small cell lung cancer patients in South Korea: A retrospective cohort study using nationwide claims data

**Yoon-Bo Shim**[1‡]**, Joo-Young Byun**[1‡]**, Ju-Yong Lee**[1,2]**, Eui-Kyung Lee**[1]**, Mi-Hai Park**[1] *

**1** School of Pharmacy, Sungkyunkwan University, Gyeonggi-do, South Korea, **2** AstraZeneca Korea, Seoul, South Korea

‡ These authors contributed equally to this work and share the first authorship on this work.
* bestway00@skku.edu

**Data Availability Statement:** To achieve Health Insurance Review and Assessment Service (HIRA) data, researchers should submit data application

## Abstract

Brain metastases (BM) are common in patients with non-small cell lung cancer (NSCLC). However, the pure economic burden of BM is unknown. This study aimed to evaluate the impact of BM on healthcare costs and resource utilization in patients with NSCLC by comparing patients with and without BM. This was a retrospective cohort analysis of South Korean health insurance review and assessment claims data. Patients with stage IIIB or IV NSCLC were identified (March 1, 2013 to February 28, 2018). We compared their two-year and per-patient-per-month (PPPM) healthcare costs and resource utilization with 1:3 propensity score-matched patients without the condition. A generalized linear model was used to estimate the impact of BM and other covariates on healthcare costs. After propensity score matching with the 33 402 newly diagnosed cases of stage IIIB or IV NSCLC, 3435 and 10 305 patients were classified as having or not having BM, respectively. Mean healthcare costs were significantly greater in patients with BM for both the two years (US$ 44 692 vs. US$ 32 230, p < .0001) and PPPM (US$ 3510 vs. US$ 2573, p < .0001). The length of hospital stay was longer in patients with BM (79.15 vs. 69.41 days for two years, p < .0001; 7.69 vs. 6.86 days PPPM, p < .0001), and patients with BM had more outpatient visits (50.61 vs. 46.43 times for two years, p < .0001; 3.64 vs. 3.40 times PPPM costs, p < .0001). The costs of drugs, radiology/radiotherapy, and admission comprised the majority of PPPM costs and were higher in patients with BM. The generalized linear model analysis suggested that patients with BM had significantly increased healthcare costs (by 1.29-fold, 95% confidence interval 1.26–1.32). BM is a significant economic burden for patients with NSCLC. Therefore, it is important to prevent BM in patients with NSCLC to reduce their economic burden.

## Introduction

Lung cancer is common cancer that has the second highest annual incidence rate among cancers. An estimated 2.2 million new patients were diagnosed with lung cancer in 2020

including research proposal and pledge of data security maintenance. After HIRA approves the application, the requested data is retrieved by HIRA from data warehouse and HIRA uploads the data to the HIRA system. Only approved researchers are permitted to access the HIRA system through permissioned remote access or designated HIRA datacenter's computer for a limited period. Since obtaining the raw data from the HIRA system is prohibited because of potentially identifying information, the final results of analysis are only obtainable. Therefore, the minimal data set could not be offered by the researchers. Healthcare Bigdata Hub HIRA homepage (https://opendata. hira.or.kr/or/orb/useGdInfo.do) provides the process of application and necessary documents.

**Funding:** This study was funded by AstraZeneca, Korea.

**Competing interests:** This study was funded by AstraZeneca, Korea and this does not alter our adherence to PLOS ONE policies on sharing data and materials. Yoon-Bo Shim, Joo-Young Byun, Eui-Kyung Lee, and Mihai Park declare that they have no conflict of interest. Ju-Yong Lee is an employee of AstraZeneca, Korea.

worldwide. Lung cancer is also the leading cause of cancer-related deaths, resulting in 1.8 million deaths per year worldwide [1]. Non-small cell lung cancer (NSCLC) is the most common type of lung cancer (80–90%) [2]. It is characterized by a rapid increase in tumor growth and aggressive spread owing to its asymptomatic nature, which leads to delays in diagnosis [3–5]. Approximately 70% of patients with NSCLC are diagnosed at an advanced stage (IIIB or IV) [6,7] and the disease severity of metastatic NSCLC is considerable, with a 5-year survival rate of less than 5% in the US. The central nervous system (CNS) is one of the most common metastatic sites of NSCLC (40%). Several studies have reported that brain metastases (BM) occur in > 10% of NSCLC patients during disease progression [8–11]. In South Korea, 10.9% of advanced-stage NSCLC patients develop BM within two years of initial diagnosis [12].

NSCLC poses a heavy economic burden on society, especially in terms of healthcare resources [13]. BM constitutes this burden because of the additional treatment required for brain lesions. Stereotactic radiosurgery, whole-brain radiation therapy, operational resection, and mutation-targeting therapies (e.g., tyrosine kinase inhibitors and immune checkpoint inhibitors) have been suggested as treatments for BM in NSCLC [9]. Previous studies have evaluated healthcare costs in patients with lung cancer and BM through retrospective analysis of healthcare claims data. A US study observed a large increase in healthcare costs in patients with epidermal growth factor receptor (EGFR)-mutated NSCLC and BM from 2012 to 2015, compared with patients with metastases at other sites [14]. Studies have shown the impact of BM on the increase in economic burden either by comparing patients before and after their diagnosis of BM [15,16] or by comparing patients with BM and patients with metastases at other sites [14]. However, these previous studies [14–16] have used designs that include not only the cost of BM but also the increasing cost of disease progression, thereby leaving the pure economic burden of BM compared with the economic cost in the absence of BM not studied. Comparing the healthcare costs of patients with and without BM would show the pure economic impact of BM with increased costs due to disease progression in both groups.

Therefore, this study aimed to assess the economic burden of BM in patients with NSCLC between stage IIIB to IV, comparing matched patients with BM (BM) and without BM (non-BM) using nationwide claims data from South Korea.

## Materials and methods

### Study design

This was a retrospective cohort study using claims data from the Health Insurance Review & Assessment (HIRA) Service from March 1, 2012, to February 29, 2020. The HIRA database is a national-level administrative database that includes information on all individuals who use insurance-covered healthcare resources in South Korea. Since all hospitals and pharmacies are required to submit procedure codes, diagnostic codes, or drug codes for every practice to be reimbursed for insurance-covered resource utilization, the information available from the HIRA database ranges from basic demographic information to details on healthcare resource utilization.

The date of the first diagnosis of lung cancer, based on the International Classification of Diseases, 10th Revision, Clinical Modification (ICD-10-CM) code C34 (malignant neoplasm of the bronchus and lung), during the index period (March 1, 2013, to February 28, 2018) was defined as the index date. We followed the patients from the index date until the last claim for lung cancer or metastatic cancers within two years after the index date. The study design is illustrated in Fig 1. This study was approved by the institutional review board of Sungkyunkwan University (IRB No. SKKU202005001).

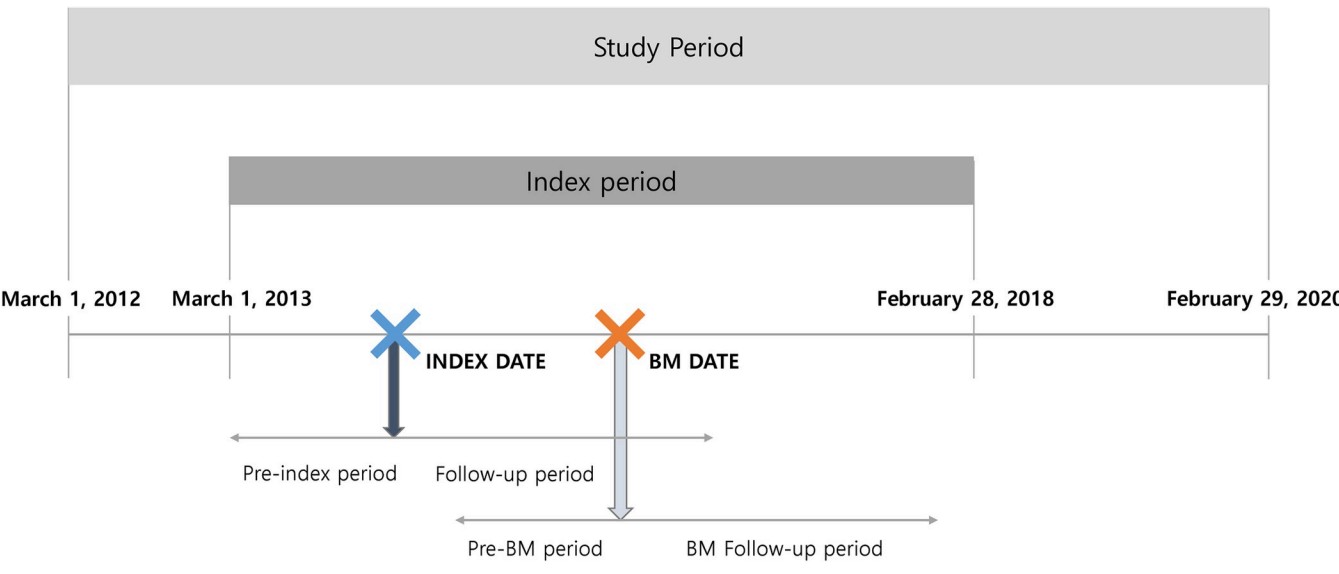

**Fig 1. Study design.** BM, brain metastases.

## Patient selection criteria

Patients with at least one inpatient or two outpatient claims for lung cancer during the index period were identified. After defining patients with lung cancer, we used additional definitions with treatments covered for NSCLC during the index period to include patients with stage IIIB or IV NSCLC. Patients who fulfilled at least one of the following three criteria were considered to have stage IIIB or IV NSCLC: (1) had been treated with afatinib, ceritinib, crizotinib, erlotinib, alectinib, or gefitinib for the first time within the first six months of the index date; (2) had undergone cytotoxic chemotherapy using paclitaxel, pemetrexed, gemcitabine, irinotecan, or etoposide at least 180 days after radiotherapy or lung surgery; or (3) had undergone cytotoxic chemotherapy for the first time without a history of radiotherapy or lung surgery [17].

To identify patients who were newly diagnosed with stage IIIB or IV NSCLC, those who had claims of lung cancer during the pre-index period (1 year before the index date) were excluded. Additionally, patients with other types of cancers during the pre-index period, including BM, were excluded to collate only the information on stage IIIB or IV NSCLC. Patients who died or had missing data within 90 days of the index date were also excluded.

## Definition of BM and matching

Among patients who were newly diagnosed with stage IIIB or IV NSCLC, those with at least two claims with ICD-10-CM code C793 (secondary malignant neoplasm of the brain and cerebral meninges) during the follow-up period were classified as having BM. For patients with BM, the date of first BM diagnosis during the index period was defined as the BM date. Therefore, patients with BM were followed-up from the BM date to the date of the last claim for lung cancer or metastatic cancer within two years after the BM date. To classify BM patients more accurately, they needed to satisfy at least one of the following conditions: (1) having at least two claims for brain imaging (magnetic resonance imaging or computed tomography) or (2) having claims for brain operation or brain radiation therapy. Patients were excluded from the BM group if they experienced cerebrovascular events during the pre-BM period (one year before the first BM date) to exclude those who underwent brain imaging or surgery for such events [17]. Patients were also excluded if they had <90 days follow-up after the BM date.

After classifying the patients with and without BM, we calculated the PS with logistic regression using age, sex, non-cancer-related Charlson comorbidity index (CCI), death, other cancer, and follow-up days as covariates. Nearest-neighbor PS matching was used to match each patient with BM with three patients without BM, based on the closest PS without replacement [18].

## Study measures

**Demographic and clinical characteristics.** To compare the demographic and clinical characteristics of the patients with and without BM before and after PS matching, we estimated the distribution of patients' sex, age, non-cancer-related CCI, death, other cancers, and follow-up days for each group. We included these covariates based on previous retrospective studies that examined the economic impact of BM in NSCLC using claims data [19,20]. Data on sex and age reflected the information recorded on the index date for patients without BM and on the BM date for patients with BM. The non-cancer-related CCI was calculated using information from the pre-index period for patients without BM and the pre-BM period for patients with BM.

We excluded patients who had cancer claims during the pre-index period. Thus, for patients with BM, the pre-index period did not include any claims for cancer. However, it was difficult to rule out the possibility that their pre-BM period included claims for cancer, especially lung cancer. Therefore, there could have been unavoidable differences in CCI between the two groups. To adjust for potential differences, we used non-cancer-related CCI by summing the scores without cancer-related items [19].

The covariate "death" indicated that the subject died with codes for death during the study period (from the index date to February 29, 2020) according to insurance data. When the codes for death were absent but the patient did not have any claims recorded for six months, the patient was considered to have died [17]. The covariate "other cancer" referred to the existence of claims for cancers other than lung cancer or BM during the follow-up period.

**Healthcare costs and healthcare resource utilization.** We analyzed the healthcare costs and healthcare resource utilization of matched patients to investigate the impact of BM and potential covariates on these two outcomes. The costs and utilization of lung cancer and BM were selected for quantification. Healthcare costs included the costs of drugs, dispensing, admission, radiology/radiotherapy, tests, procedures, consultations, and others. We compared the mean number of hospitalizations, length of hospital stay, and the number of outpatient visits between the patients with and without BM. In addition, the proportion of patients who had been hospitalized or had outpatient visits was also compared between the two groups. Costs and utilization were analyzed for up to two years, and we presented the total cost for two years and the cost per patient per month (PPPM). All costs were converted from South Korean won to 2019 US dollars using the exchange rate (1166 South Korean won = 1 US dollar) obtained from the Korean Statistical Information Service.

## Statistical analysis

The basic demographic and clinical characteristics of the selected patients, considering their BM status, are presented descriptively. We used means and standard deviations for continuous variables and frequencies and percentages for categorical variables. To determine the statistical significance of differences between patients with and without BM, t-tests and chi-square tests were used for continuous and categorical variables, respectively. The presented p-value was based on a significance level of 5% in the two-sided condition. To estimate the impact of each covariate on healthcare costs, we used a generalized linear model (GLM) with a log link and

gamma distribution. The results of the GLM analysis are reported in an exponential form with 95% confidence intervals (CIs). All analyses were conducted using SAS Enterprise 6.1 (SAS Institute Inc., Cary, NC, US).

## Results

### Demographic and clinical characteristics of patients

In total, 214 555 patients had claims for lung cancer during the index period. Among these, 33 402 patients fulfilled the inclusion criteria for newly diagnosed stage IIIB or IV NSCLC. After PS matching, 3435 and 10 305 patients were classified as having BM and non-BM, respectively. Fig 2 illustrates the patient selection process. Before matching, there were statistically significant differences between patients with and without BM in terms of sex, age, non-cancer-related CCI, death, other cancers, and follow-up duration (p < .0001 in the *t*-test or chi-squared test). However, after matching, the characteristics were well-balanced between the two groups. More than 70% of the patients were men. The average age of the matched cohorts was approximately 64 years, with the highest number being aged 65–79 years, followed by 50–64 years. The mean non-cancer-related CCI was approximately 1.94, and among the four categories of CCI, the value of 1 was the largest, followed by 3 or more. In terms of death, approximately 63% of patients died during the study period. Approximately 69% of the patients had claims of other types of cancer during the follow-up period, with a mean of 448 days. Table 1 presents the demographic and clinical characteristics of all the patients with and without BM.

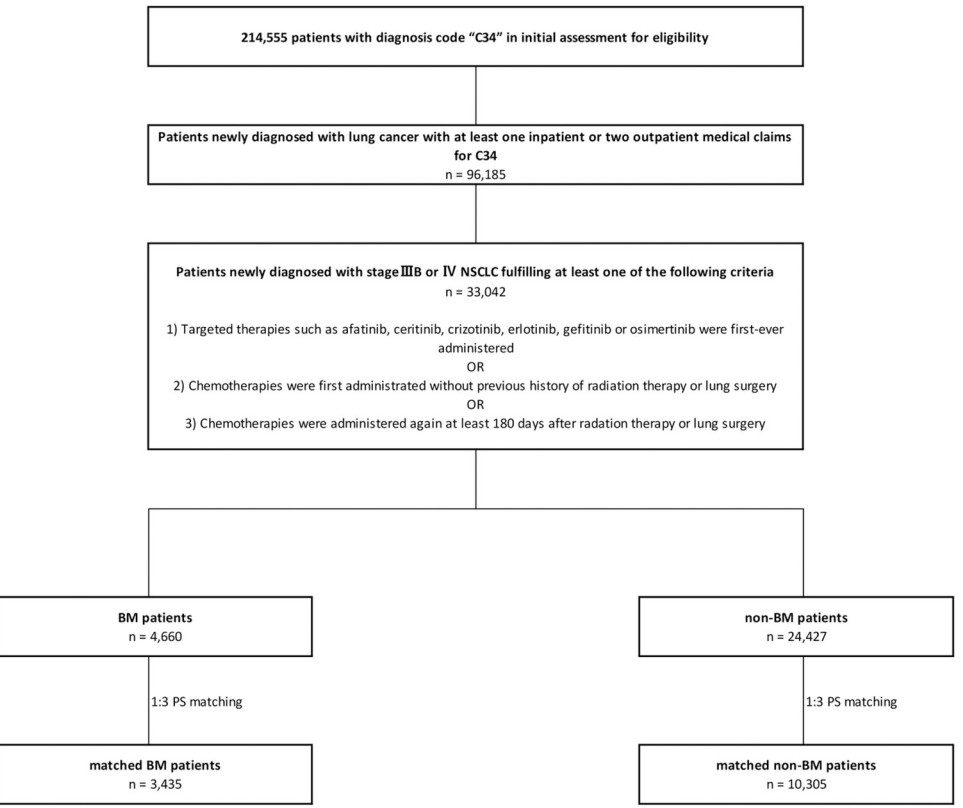

**Fig 2. Flowchart of the study patients.** NSCLC, non-small cell lung cancer; BM, brain metastases; PS, propensity score.

**Table 1. Demographic and clinical characteristics of patients with and without brain metastases.**

|  | Unmatched | | | | Matched | | | |
|---|---|---|---|---|---|---|---|---|
|  | **All** | **BM** | **Non-BM** | **p-value** | **All** | **BM** | **Non-BM** | **p-value** |
| All | 29 087 (100.00) | 4660 (100.00) | 24 427 (100.00) |  | 13 740 (100.00) | 3435 (100.00) | 10 305 (100.00) |  |
| Sex, n (%) |  |  |  | <0.0001 |  |  |  | 0.6641 |
| Male | 21 265 (73.11) | 3012 (64.64) | 18 253 (74.72) |  | 9756 (71.00) | 2449 (71.30) | 7307 (70.91) |  |
| Female | 7822 (26.89) | 1648 (35.36) | 6174 (25.28) |  | 3984 (29.00) | 986 (28.70) | 2998 (29.09) |  |
| Age, mean (SD) | 65.4 (10.15) | 61.89 (10.31) | 66.07 (9.98) | <0.0001 | 64.22 (9.67) | 64.14 (9.77) | 64.25 (9.64) | 0.3357 |
| Age group, n (%) |  |  |  | <0.0001 |  |  |  | 0.9349 |
| <50 | 1951 (6.71) | 518 (11.12) | 1443 (5.87) |  | 990 (7.21) | 254 (7.39) | 736 (7.14) |  |
| 50–64 | 10 690 (36.75) | 2213 (47.49) | 8477 (34.7) |  | 5660 (41.19) | 1419 (41.31) | 4241 (41.15) |  |
| 65–79 | 14 699 (50.53) | 1791 (38.43) | 12 908 (52.84) |  | 6559 (47.74) | 1627 (47.37) | 4932 (47.86) |  |
| >80 | 1747 (6.01) | 138 (2.96) | 1609 (6.59) |  | 531 (3.86) | 135 (3.93) | 396 (3.84) |  |
| Non-cancer-related CCI, mean (SD) | 2.02 (1.75) | 1.86 (1.55) | 2.06 (1.78) | <0.0001 | 1.94 (1.64) | 1.93 (1.64) | 1.94 (1.64) | 0.9252 |
| CCI group, n (%) |  |  |  | <0.0001 |  |  |  | 0.9783 |
| 0 | 4874 (24.33) | 834 (17.90) | 4558 (18.66) |  | 2534 (18.44) | 642 (18.69) | 1892 (18.36) |  |
| 1 | 7077 (19.87) | 1428 (30.64) | 6737 (27.58) |  | 4042 (29.42) | 1009 (29.37) | 3033 (29.43) |  |
| 2 | 5780 (39.04) | 1154 (24.76) | 5197 (21.28) |  | 3149 (22.92) | 783 (22.79) | 2366 (22.96) |  |
| 3+ | 11 356 (39.04) | 1244 (26.70) | 7935 (32.48) |  | 4015 (29.22) | 1001 (29.14) | 3014 (29.25) |  |
| Death, n (%) |  |  |  | <0.0001 |  |  |  | 0.1936 |
| Yes | 15 978 (54.93) | 3401 (72.98) | 12 577 (51.49) |  | 8737 (63.59) | 2216 (64.51) | 6521 (63.28) |  |
| No | 13 109 (45.07) | 1259 (27.02) | 11 850 (48.51) |  | 5003 (36.41) | 1219 (35.49) | 3784 (36.72) |  |
| Other cancer, n (%) |  |  |  | <0.0001 |  |  |  | 0.9660 |
| Yes | 18 616 (64.00) | 3416 (73.3) | 15 200 (62.23) |  | 9468 (68.91) | 2368 (68.94) | 7100 (68.9) |  |
| No | 10 471 (36.00) | 1244 (26.7) | 9227 (37.77) |  | 4272 (31.09) | 1067 (31.06) | 3205 (31.1) |  |
| Follow-up days, mean (SD) | 488.86 (236.51) | 411.00 (229.66) | 503.71 (234.88) | <0.0001 | 448.34 (237.34) | 443.47 (237.77) | 449.96 (237.18) | 0.8554 |

BM, brain metastases; CCI, Charlson comorbidity index; SD, standard deviation.

## Healthcare costs and resource utilization

As shown in Table 2, the mean healthcare costs were significantly greater in patients with BM than in patients without BM for both the two years (US$ 44 692 vs. 32 230, p < .0001) and the PPPM (US$ 3510 vs. 2753, p < .0001). In addition, the healthcare costs of both inpatients and outpatients were higher in the BM group than in the non-BM group for the two years and PPPM (p < .0001). For inpatients, admission and radiology/radiotherapy accounted for the largest proportion, and the costs of these items were higher in patients with BM than in those without BM (p < .0001). For outpatients, the costs of drugs, radiology/radiotherapy, and consultations were higher than those of other items, and the costs were higher in patients with BM than in those without BM (p < .0001). Fig 3 shows the PPPM costs for each item for both inpatients and outpatients. The PPPM costs of drugs, radiology/radiotherapy, admission, and consultations were higher in patients with BM than in those without BM (p < .0001). In both groups, the PPPM costs of drugs were the highest among the various cost items, followed by radiology/radiotherapy and admission.

The PPPM number of hospitalizations was slightly lower in patients with BM than in those without BM, and the difference for two years was not significant (5.99 vs. 6.17, p = .0669 for two years; 0.56 vs. 0.59, p = .0009 for PPPM). The mean length of hospital stay was higher in patients with BM (79.15 vs. 69.41 days, p < .0001 for two years; 7.69 vs. 6.86 days, p < .0001 for PPPM). Further, patients with BM had more outpatient visits than patients without BM (50.61 vs. 46.43 times, p < .0001 for two years; 3.64 vs. 3.40 times, p < .0001 for PPPM). The

**Table 2. Two-year and PPPM healthcare costs and resource utilization in matched patients with and without brain metastases.**

| | 2-year Healthcare Costs & Resource Utilization | | | | PPPM Healthcare Costs & Resource Utilization | | | |
|---|---|---|---|---|---|---|---|---|
| | **All** | **BM** | **Non-BM** | **p-value** | **All** | **BM** | **Non-BM** | **p-value** |
| **Medical costs** | | | | | | | | |
| **Total costs** | 35 353 | 44 692 | 32 230 | <0.0001 | 2942 | 3510 | 2753 | <0.0001 |
| **Inpatient subtotal** | 18 838 | 21 368 | 18 009 | <0.0001 | 1897 | 2135 | 1820 | <0.0001 |
| Drugs | 3488 | 3631 | 3442 | 0.2839 | 356 | 353 | 357 | 0.2717 |
| Dispensing | 797 | 807 | 793 | 0.9291 | 91 | 91 | 90 | 0.7043 |
| Admission | 4172 | 4610 | 4028 | <0.0001 | 431 | 476 | 416 | <0.0001 |
| Radiology/radiotherapy | 3489 | 5645 | 2783 | <0.0001 | 359 | 548 | 297 | <0.0001 |
| Tests | 2655 | 2296 | 2773 | <0.0001 | 281 | 240 | 295 | <0.0001 |
| Procedures | 2333 | 1550 | 2590 | <0.0001 | 202 | 150 | 219 | <0.0001 |
| Consultations | 537 | 657 | 497 | <0.0001 | 54 | 67 | 50 | <0.0001 |
| Other | 1366 | 2171 | 1103 | <0.0001 | 124 | 210 | 96 | <0.0001 |
| **Outpatient subtotal** | 16 976 | 24 162 | 14 568 | <0.0001 | 1087 | 1454 | 964 | <0.0001 |
| Drugs | 8836 | 13 365 | 7318 | <0.0001 | 538 | 754 | 465 | <0.0001 |
| Dispensing | 176 | 170 | 178 | 0.1685 | 13 | 13 | 14 | 0.0971 |
| Radiology/radiotherapy | 3892 | 4785 | 3592 | <0.0001 | 285 | 345 | 265 | <0.0001 |
| Tests | 647 | 621 | 656 | 0.002 | 48 | 45 | 50 | <0.0001 |
| Procedures | 80 | 70 | 84 | 0.0065 | 6 | 6 | 7 | 0.0792 |
| Consultations | 3137 | 4858 | 2560 | <0.0001 | 182 | 273 | 151 | <0.0001 |
| Other | 208 | 293 | 180 | <0.0001 | 14 | 18 | 13 | <0.0001 |
| **Resource Utilization** | | | | | | | | |
| **Proportion of patients hospitalized, n (%)** | 13 468 (98.24) | 3324 (96.77) | 10 144 (98.73) | < .0001 | 13 468 (98.24) | 3324 (96.77) | 10 144 (98.73) | < .0001 |
| Number of hospitalizations | 6.13 | 5.99 | 6.17 | 0.0669 | 0.58 | 0.56 | 0.59 | 0.0009 |
| Length of hospital stay (days) | 71.85 | 79.15 | 69.41 | < .0001 | 7.07 | 7.69 | 6.86 | < .0001 |
| **Proportion of patients with outpatient visits, n (%)** | 13 604 (99.23) | 3414 (99.39) | 10 190 (99.18) | 0.2300 | 13 604 (99.23) | 3414 (99.39) | 10 190 (99.18) | 0.2300 |
| Number of outpatient visits | 47.47 | 50.61 | 46.43 | < .0001 | 3.46 | 3.64 | 3.40 | < .0001 |

BM, brain metastases; PPPM, per patient per month.

All costs were converted into US dollars from the Korean won using an exchange rate of 2019.

proportion of hospitalized patients was lower in those with BM than in those without (96.77% vs. 98.73%, p < .0001). The difference in the proportion of patients with outpatient visits between the two groups was not significant (p = .23).

## Impact of the covariates on healthcare costs

Table 3 presents the results of the GLM analysis. The results suggest that BM significantly increased PPPM healthcare costs when the other covariates were controlled for (1.292, 95% CI 1.264–1.320). Of all the covariates included in the GLM analysis, death had the greatest influence on healthcare costs (all: 1.574, 95% CI 1.535–1.613; non-BM: 1.677, 95% CI 1.629–1.726; BM: 1.321, 95% CI 1.261–1.383). The impact of sex on healthcare costs was insignificant for all patients. However, for patients with BM, healthcare costs were 1.087 times higher in female patients compared with in men (95% CI 1.040–1.136). Compared with patients aged 80 years and older, younger patients tended to have significantly higher healthcare costs, except for patients with BM aged between 65 and 79 years (1.095, 95% CI .996–1.203). Regarding non-cancer-related CCI, the 95% CI of most results implied the insignificance of the estimated results. However, in the group with a non-cancer-related CCI ≥ three or more, the results

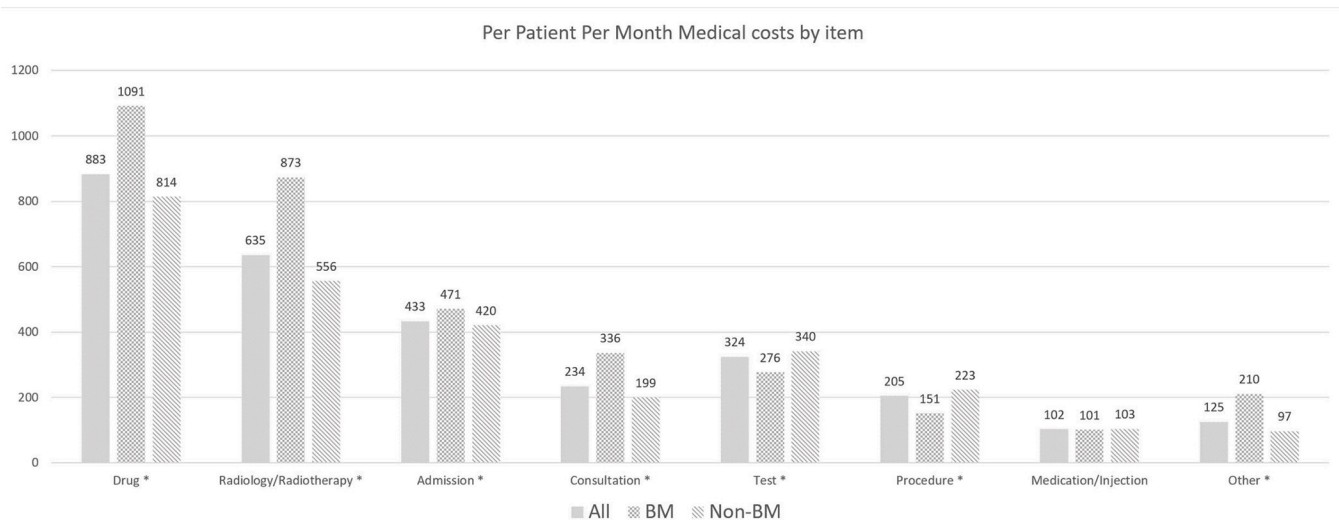

**Fig 3. PPPM healthcare costs by item in patients with and without brain metastases.** All costs were converted into US dollars from the Korean won using the exchange rate of 2019. The statistical significance of the difference between the costs of patients with and without BM was assessed using the *t*-test. The asterisk (*) indicates p<0.05; PPPM, per patient per month; BM, brain metastases.

were statistically significant in all patients and patients without BM. Patients who had claims for other types of cancers during follow-up had higher healthcare costs than those who did not (all: 1.081, 95% CI 1.059–1.103; non-BM, 1.088; 95% CI, 1.063–1.114; BM, 1.071; 95% CI, 1.031–1.114).

## Discussion

This study used nationwide population-based claims data to estimate the economic burden of BM by comparing patients with stage IIIB or IV NSCLC with and without BM. For the two

**Table 3. Generalized linear model results indicating the impact of each covariate on PPPM healthcare cost.**

|  | All | BM | Non-BM |
|---|---|---|---|
| **BM** | 1.292 (1.264–1.320) [a] | - | - |
| **Sex** | 1.018 (0.995–1.042) | 1.087 (1.040–1.136) [a] | 1.000 (0.974–1.027) |
| **Age group** |  |  |  |
| <50 | 1.378 (1.297–1.465) [a] | 1.321 (1.178–1.480)[a] | 1.410 (1.313–1.514) [a] |
| 50–64 | 1.251 (1.188–1.317) [a] | 1.195 (1.086–1.316)[a] | 1.279 (1.205–1.358) [a] |
| 65–79 | 1.153 (1.096–1.212) [a] | 1.095 (0.996–1.203) | 1.177 (1.110–1.248) [a] |
| >80 | Reference | Reference | Reference |
| **Non-cancer-related CCI group** |  |  |  |
| 0 | Reference | Reference | Reference |
| 1 | 1.017 (0.990–1.046) | 0.986 (0.935–1.039) | 1.031 (0.999–1.065) |
| 2 | 1.004 (0.974–1.034) | 0.986 (0.932–1.043) | 1.012 (0.978–1.047) |
| 3+ | 1.044 (1.014–1.074) [a] | 0.975 (0.923–1.029) | 1.066 (1.031–1.102) [a] |
| **Death** | 1.574 (1.535–1.613) [a] | 1.321 (1.261–1.383) [a] | 1.677 (1.629–1.726) [a] |
| **Other cancer** | 1.081 (1.059–1.103) [a] | 1.071 (1.031–1.114)[a] | 1.088 (1.063–1.114) [a] |

All values were expressed in an exponential form (beta coefficient). PPPM, per patient per month; BM, brain metastases; CCI, Charlson Comorbidity Index.
[a]Statistically significant 95% confidence interval.

years, the mean healthcare costs of BM patients increased by US$ 12 658 compared with that of non-BM patients (US$ 45 580 vs. US$ 32 922). The increase for BM patients was about four times the health expenditure per capita in Korea in 2018 (US$ 3192) [21]. The PPPM costs for patients with NSCLC and BM increased by 1.292-fold compared to PPPM costs for patients without BM.

Previous studies have also shown the economic burden of BM. Fernandes et al., in a study of patients with NSCLC using epidermal growth factor receptor tyrosine kinase inhibitors, reported that the healthcare cost after BM diagnosis increased significantly compared with that before BM diagnosis (US$ 9670 vs. US$ 17 581) [14]. Similarly, the monthly cost per patient was US$ 5983 before the diagnosis of BM but increased markedly to US$ 22 645 after BM in a US study based on patients treated with crizotinib [16]. The cost difference observed in these studies was greater than that observed in our study. Since healthcare costs increase as disease severity worsens [22,23], before-and-after comparisons used in previous studies might have included costs due to disease progression in the costs after diagnosis. Since we ruled out the possibility of cost increase due to disease progression by comparing patients with and without BM, our result would represent the economic burden of BM itself.

In our study, PPPM costs in patients with BM comprised more inpatient costs than outpatient costs. This is consistent with previous studies that showed more PPPM costs in inpatient settings than in outpatient settings in patients with BM [19,20]. Moreover, as medications and radiation therapy have been suggested as the main cost drivers in patients with BM [19], drug costs comprised the greatest part of the total costs, followed by radiology/radiotherapy in our study. These greatest cost components were incurred in both inpatient and outpatient settings, with drug costs incurred mostly in outpatient settings and radiology/radiotherapy costs incurred mostly in inpatient settings.

Longer PPPM length of hospital stay (3.0 vs. 1.2 days) and more frequent outpatient visits (1.7 vs. 1.4 times) in anaplastic lymphoma kinase (ALK) inhibitor-treated NSCLC patients with BM compared to those without BM were also observed in a previous study [19]. Compared to the study, our results showed a higher PPPM length of hospital stay and outpatient visits in patients with and without BM. This could be caused by the clinical circumstances of South Korea, in which relatively more healthcare resources are used than in other countries under fee-for-service systems [21].

The GLM results in our study are comparable to those of previous studies in the US. NSCLC patients who previously received crizotinib and ALK inhibitor treatment as second-line treatment had an average PPPM cost 1.37-fold higher than those without BM [19]. Another study analyzed claims data from the US to estimate the effect of alectinib on preventing BM in patients with ALK-positive NSCLC and found that ALK-positive NSCLC patients with BM accounted for approximately 1.32-fold higher PPPM costs than patients without BM [20]. These results are similar to the 1.292-fold increase observed in the present study. Since the two previous studies were limited to ALK inhibitor users, our study results would be more generalizable to BM in NSCLC since we used a broader scope of patient selection criteria, including the patient groups of previous studies. Consistent with a previous study that indicated that younger lung cancer patients aged less than 65 years had higher medical costs compared to patients aged $\geq$ 65 years [24], our GLM analysis showed that younger patients incur higher healthcare costs. It could be inferred that higher costs were incurred because younger age has been reported to be associated with more tolerable and aggressive cancer treatments [25,26]. Likewise, higher costs of targeted therapy, ICI, and radiology/radiotherapy were incurred in younger patients in our study.

This study has several strengths. First, it used the HIRA database, which includes claims data of the entire South Korean population. Therefore, this study included all patients with

stage IIIB or IV NSCLC in South Korea and provided representative results. In addition, the national health insurance system is a fee-for-service system, even for inpatient services [27]. In this study, the drugs and treatments received during hospitalization were confirmed and a hospitalization-related subgroup analysis was performed. Second, this study was designed to measure the pure impact of BM. The effect of lung cancer progression cannot be excluded when comparing costs before and after BM diagnosis. To compensate for this, patients without BM were used as the control group and compared with patients with BM. To ensure that all patients had similar disease severity, we selected only patients with stage IIIB or IV NSCLC. In addition, PS matching was performed to eliminate the difference in basic characteristics between patients with and without BM, and healthcare resource utilization and healthcare costs were measured after the occurrence of BM. GLM analysis was performed to adjust for factors that might have affected these costs. Through this study design, we were able to measure the pure impact of BM in patients with NSCLC.

This study has several limitations. First, the HIRA claims data are recorded as reimbursed services only. Therefore, drugs and treatments not covered by insurance were not identified in this study. Second, there was a lack of information on the characteristics of tumors in the HIRA data. No information on the disease progression or clinical stage of cancer was available. In addition, the ICD-10 codes, the diagnostic codes from the HIRA database, do not distinguish between NSCLC and small cell lung cancer. To improve the accuracy in defining patients with NSCLC, we consulted clinicians on the relevant diagnostic and procedure codes for NSCLC and used drugs and treatments exclusively for patients with NSCLC [17]. Information on the diagnoses, procedures, and drugs used might have been inaccurate due to misclassification; however, a previous validation study that compared HIRA data and actual medical records of hospitalized patients reported a concordance of 82% [28]. Lastly, although our study results highly represent the economic burden of NSCLC and BM patients in South Korea using HIRA claims data, differences between countries should be considered when generalizing our results to the global level.

## Conclusion

BM significantly increases healthcare costs for patients with stage IIIB or IV NSCLC. Therefore, the prevention of BM in these patients is needed to reduce the economic burden.

## Acknowledgments

We would like to thank Editage (www.editage.co.kr) for English language editing.

## Author Contributions

**Conceptualization:** Yoon-Bo Shim, Joo-Young Byun, Ju-Yong Lee, Mi-Hai Park.

**Data curation:** Yoon-Bo Shim.

**Formal analysis:** Yoon-Bo Shim, Joo-Young Byun.

**Funding acquisition:** Ju-Yong Lee.

**Investigation:** Yoon-Bo Shim, Joo-Young Byun, Ju-Yong Lee.

**Methodology:** Yoon-Bo Shim, Joo-Young Byun, Eui-Kyung Lee, Mi-Hai Park.

**Project administration:** Eui-Kyung Lee, Mi-Hai Park.

**Resources:** Eui-Kyung Lee, Mi-Hai Park.

**Software:** Yoon-Bo Shim, Joo-Young Byun.

**Supervision:** Eui-Kyung Lee, Mi-Hai Park.

**Validation:** Yoon-Bo Shim, Joo-Young Byun, Ju-Yong Lee, Eui-Kyung Lee, Mi-Hai Park.

**Visualization:** Yoon-Bo Shim, Joo-Young Byun.

**Writing – original draft:** Yoon-Bo Shim, Joo-Young Byun, Ju-Yong Lee.

**Writing – review & editing:** Yoon-Bo Shim, Joo-Young Byun, Ju-Yong Lee, Eui-Kyung Lee, Mi-Hai Park.

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
