## [Decision Letter · Decision Letter 0]

20 Jun 2022

PONE-D-21-30124Economic burden of brain metastases in non-small cell lung cancer patients in South Korea: A retrospective cohort study using nationwide claims dataPLOS ONE

Dear Dr. Park,

Thank you for submitting your manuscript to PLOS ONE. After careful consideration, we feel that it has merit but does not fully meet PLOS ONE’s publication criteria as it currently stands. Therefore, we invite you to submit a revised version of the manuscript that addresses the points raised during the review process.

Your manuscript has undergone the peer-review process and the reviewers have provided their comments/suggestions. Kindly address these points/concerns before we make a decision. 

We look forward to receiving your revised manuscript.

Kind regards,

Kingston Rajiah

Academic Editor

PLOS ONE

Journal Requirements:

2. Thank you for stating the following in the Financial Disclosure section: "This study was funded by AstraZeneca, Korea."

We note that you received funding from a commercial source: AstraZeneca

Reviewers' comments:

Reviewer's Responses to Questions

**Comments to the Author**

1. Is the manuscript technically sound, and do the data support the conclusions?

Reviewer #1: Yes

Reviewer #2: Yes

Reviewer #3: Yes

2. Has the statistical analysis been performed appropriately and rigorously? 

Reviewer #1: Yes

Reviewer #2: Yes

Reviewer #3: Yes

3. Have the authors made all data underlying the findings in their manuscript fully available?

Reviewer #1: Yes

Reviewer #2: Yes

Reviewer #3: Yes

4. Is the manuscript presented in an intelligible fashion and written in standard English?

Reviewer #1: Yes

Reviewer #2: Yes

Reviewer #3: Yes

5. Review Comments to the Author

Reviewer #1: This manuscript identifies an economic burden of brain metastases in non-small cell lung cancer patients in South Korea using nationwide claims data. This study is timely considering the trend in increasing prevalence of non-small cell lung cancer patients in nationwide. So, this study will be good addition to the literature for the future reader. However, the manuscript will need to develop and require more revision based on my comments.

The authors must provide the rationale for selection of covariates based on either theory-driven or literature including categorization of your independent variables.

This manuscript used the South Korea’s claims data. However, I didn’t see any epidemiological information in South Korea in the introduction. Please add one paragraph of the information about South Korea.

The authors provided USD monetary unit in the paper. Did you convert monetary unit from South Korea to USD in the paper? If so, please indicate this information in the manuscript.

The policy implication of this study can be further addressed; the discussion section should be further developed specially limitation section (i.e., generalizability issue).

For the quality of written English: Acceptable level, but overall, careful proof reading by an expert English writer would assist identifying and revision the article

The methods section is very clear and provides rationale and evidence for each of the data sources used. The analysis is clearly described.

Otherwise, well-written manuscript.

Level of interest: An article of importance in its field!!

Reviewer #2: The study on the economic burden of brain metastases in NSCLC in South Korea is very interesting. It’s a well-designed study, however, I have some minor comments that the authors may want to address.

Minor comments

• Abstract (page 2, line 22): Adding 1-2 sentence(s) about the background/rationale of this study will give your audience better perspective about its importance.

• Abstract (page 2, line 34): The authors used the term “hospitalization days” throughout the manuscript. This language does not match with today’s literature. More appropriate language would be “length of hospital stay.”

• Abstract (page 2, line 38): The term “principal components” makes the description confusing because the results didn’t come from principal components analysis or unsupervised learning. I would suggest using other terms for GLM or what the authors used.

• Introduction (page 3, lines 47-48): please add a reference.

• Introduction (page 3, lines 54-56): If possible, it would be better to have worldwide data and Korean data rather than the U.S and Korea. This would make it consistent with previous information, as well as show the current situation in Korea.

• Introduction (page 3, lines 59-60): Talking about the QOL seems out of the scope of this paper.

• Introduction (page 3, lines 47-60): If possible, it would be better to have information regarding Korea rather than the U.S.

• Introduction (page 4, line 67): For reference 14, please specify which county’s results.

• Introduction (page 4, line 67): “using US claims data” – The authors used “United States” previously. Please be consistent.

• Introduction (page 4, lines 69-72): “Studies have also ….” Please add references.

• Introduction (page 4, lines 69-72): “However, previous studies have also ….” Please add references.

• Methods: The words “code C34” and “code C793” were repeated many times. Once a term has been defined, it does not need to be defined again. Deleting duplicate descriptions will aid readability and decrease unnecessary word use.

• Methods (page 8, line 162): “...radiology/radiotherapy, and other costs.” Please specify what types of costs were included in the other category.

• Discussion (page 19, lines 336-344): Additional limitations to consider include generalizability beyond the study sample, and the lack of information about the characteristics of the tumor.

Reviewer #3: 1. What is the unit of 3.64 vs. 3.40 in the abstract (page 2, page 37)

2. In the abstract, the author said “It is important to prevent BM in patients with NSCLC to reduce the economic burden on society.” But this study only estimated the medical cost from the healthcare system perspective, not societal perspective. Is this interpretation appropriate?

3. Minor: page3, line 61, please use the abbreviations NSCLC and BM, because the authors already spelled out and abbreviated those words.

4. Page 5, line 93: What is the meaning of “From the index date, patients were followed up for up to 2 years or until the date were removed”

5. Immuno-cancer drugs such as pembrolizumab and atezolizumab are also used for NSCLC, but why is there no selection criteria?

6. According to table3, the younger the age group, the higher the medical cost tends to be. Add an interpretation of this to the discussion.

7. In the introduction section, the examples of treatment options used for brain metastases (BM) in NSCLC should be explained to support increased economic burden from additional treatment needed for BM.

8. Please how you enabled presenting “the pure economic burden of BM compared with the absence of BM” unlike other previous studies in the Introduction section.

9. When referring to previous studies in the second paragraph of the Discussion section, please relate the studies with your results. The explanation about French study seems unrelated with your results.

10. Please compare the results of your study and previous studies in terms of main items that comprised larger healthcare costs in BM patients in the third paragraph of the Discussion section.

6. PLOS authors have the option to publish the peer review history of their article (what does this mean?). If published, this will include your full peer review and any attached files.

Reviewer #1: No

Reviewer #2: No

Reviewer #3: No

---

## [Author Response · Author response to Decision Letter 0]

17 Aug 2022

Thanks to thoughtful and meaningful comments from the reviewers, we believe our manuscript has been strengthened.

---

## [Editor Report · Decision Letter 1]

7 Sep 2022

Economic burden of brain metastases in non-small cell lung cancer patients in South Korea: A retrospective cohort study using nationwide claims data

PONE-D-21-30124R1

Dear Dr. Park,

We’re pleased to inform you that your manuscript has been judged scientifically suitable for publication and will be formally accepted for publication once it meets all outstanding technical requirements.

Kind regards,

Kingston Rajiah

Academic Editor

PLOS ONE
---

## [Editor Report · Acceptance letter]

11 Sep 2022

PONE-D-21-30124R1 

Economic burden of brain metastases in non-small cell lung cancer patients in South Korea: A retrospective cohort study using nationwide claims data 

Dear Dr. Park:

I'm pleased to inform you that your manuscript has been deemed suitable for publication in PLOS ONE. Congratulations! Your manuscript is now with our production department. 

Kind regards, 

on behalf of

Associate Professor Kingston Rajiah 

Academic Editor

PLOS ONE